# Entropy-Based Discovery of Summary Causal Graphs in Time Series

**DOI:** 10.3390/e24081156

**Published:** 2022-08-19

**Authors:** Charles K. Assaad, Emilie Devijver, Eric Gaussier

**Affiliations:** 1R&D Department, EasyVista, 38000 Grenoble, France; 2Department of Mathematics, Information and Communication Sciences, University of Grenoble Alpes, CNRS, Grenoble INP, LIG, 38000 Grenoble, France

**Keywords:** causal discovery, time series, summary causal graph, mutual information

## Abstract

This study addresses the problem of learning a summary causal graph on time series with potentially different sampling rates. To do so, we first propose a new causal temporal mutual information measure for time series. We then show how this measure relates to an entropy reduction principle that can be seen as a special case of the probability raising principle. We finally combine these two ingredients in PC-like and FCI-like algorithms to construct the summary causal graph. There algorithm are evaluated on several datasets, which shows both their efficacy and efficiency.

## 1. Introduction

Time series arise as soon as observations, from sensors or experiments, for example, are collected over time. They are present in various forms in many different domains, such as healthcare (through, e.g., monitoring systems), Industry 4.0 (through, e.g., predictive maintenance and industrial monitoring systems), surveillance systems (from images, acoustic signals, seismic waves, etc.), or energy management (through, e.g., energy consumption data), to name but a few. We are interested in this study in analyzing quantitative discrete-time series to detect the causal relations that exist between them under the assumption of consistency throughout time [1] which states that causal relationships remain constant in direction throughout time. Under this assumption, one can replace the infinite full-time causal graph (Figure 1a) by a window causal graph (Figure 1b) defined as follows:

**Definition** **1**(Window causal graph [1])**.**
*Let X be a multivariate discrete-time series and G=(V,E) the associated* window causal graph *for a window of size τ. The set of vertices in that graph consists of the set of components $X^{1},\ldots ,X^{d}$ at each time t,…,t+τ. The edges E of the graph are defined as follows: variables $X_{t-i}^{p}$ and $X_{t}^{q}$ are connected by a lag-specific directed link $X_{t-i}^{p}\to X_{t}^{q}$ in G pointing forward in time if and only if $X^{p}$ causes $X^{q}$ at time t with a time lag of 0≤i≤τ for p≠q and with a time lag of 0<i≤τ for p=q.*

The window causal graph only covers a fixed number of time instants, which are sufficient to understand the dynamics of the system given the largest time gap between causes and effects. This said, it is difficult for an expert to provide or validate a window causal graph because it is difficult to determine which exact time instant is the cause of another. In contrast, there exists another type of causal graph where a node corresponds to a time series, as illustrated in Figure 1c, which usually can be analyzed or validated by an expert easily [2,3,4]. Such graphs are referred to as summary causal graphs [1] and are defined as follows:

**Definition** **2**(Summary causal graph [1])**.**
*Let X be a multivariate discrete-time series and G=(V,E) the associated* summary causal graph. *The set of vertices in that graph consists of the set of time series $X^{1},\ldots ,X^{d}$. The edges E of the graph are defined as follows: variables $X^{p}$ and $X^{q}$ are connected if and only if there exist some time t and some time lag i such that $X_{t-i}^{p}$ causes $X_{t}^{q}$ at time t with a time lag of 0≤i for p≠q and with a time lag of 0<i for p=q.*

Note that a summary causal graph can be deduced from a window causal graph, but the reverse is not true. We focus in this study on the summary causal graph, as it provides a simple and efficient view on the causal relations that exist between time series. In particular, we are interested in inferring a summary causal graph without passing by a window causal graph to avoid unnecessary computations.

An important aspect of real-world time series is that different time series, as they measure different elements, usually have different sampling rates. Despite this, the algorithms that have been developed so far to discover causal structures [5,6,7,8] rely on the idealized assumptions that all time series have the same sampling rates with identical timestamps (assuming identical timestamps in the case of identical sampling rates seems reasonable as one can shift time series so that they coincide in time).

We introduce in this paper a constraint-based strategy to infer a summary causal graph from discrete-time series with continuous values with equal or different sampling rates under the two classical assumptions of causal discovery (causal Markov condition, faithfulness), in addition to acyclicity in summary causal graphs. In summary, our contribution is four-fold:First of all, we propose a new causal temporal mutual information measure defined on a window-based representation of time series;We then show how this measure relates to an entropy reduction principle, which can be seen as a special case of the probability raising principle;We also show how this measure can be used for time series with different sampling rates;We finally combine these three ingredients in PC-like and FCI-like algorithms [9] to construct the summary causal graph from time series with equal or different sampling rates.

The remainder of the paper is organized as follows: Section 2 describes the related work. Section 3 introduces the (conditional) mutual information measures we propose for time series and the entropy reduction principle that our method is based on. Section 4 presents two causal discovery algorithms we developed on top of these measures. The causal discovery algorithms we propose are illustrated and evaluated on datasets, including time series with equal and different sampling rates and a real dataset in Section 5. Finally, Section 6 concludes the paper.

## 2. Related Work

Granger Causality is one of the oldest methods proposed to detect causal relations between time series. However, in its standard form [5], it is known to handle a restricted version of causality that focuses on linear relations and causal priorities, as it assumes that the past of a cause is necessary and sufficient for optimally forecasting its effect. This approach has nevertheless been improved since then in MVGC [10] and has recently been extended to handle nonlinearities through an attention mechanism within convolutional networks [8]. This last extension is referred to as TCDF. Score-based approaches [11] search over the space of possible graphs, trying to maximize a score that reflects how well the graph fits the data. Recently, a new score-based method called Dynotears [12] was presented to infer a window causal graph from time series. In a different line, approaches based on the noise assume that the causal system can be defined by a set of equations that explains each variable by its direct causes and an additional noise. Causal relations are in this case discovered using footprints produced by the causal asymmetry in the data. For time series, the most popular nonlinear algorithm in this family is TiMINo [6], which discovers a causal relationship by looking at independence between the noise and the potential causes. The most popular approaches for inferring causal graphs are certainly constraint-based approaches, based on the PC and FCI algorithms [9], which turn knowledge about (conditional) independencies into causal knowledge assuming the causal Markov condition and faithfulness. Several algorithms, adapted from non-temporal causal graph discovery algorithms, have been proposed in this family for time series, namely oCSE by [13], which is limited to one time lag between causal relations, PCMCI [7], and tsFCI [14]. Our work is thus more closely related to that of [7,14], which use constraint-based strategies to infer a window causal graph. However, we focus here on the summary causal graph, and we introduce a new causal temporal mutual information measure and entropy reduction principle on which we ground our causal discovery algorithms. Constraint-based methods have been also used jointly with other approaches (e.g., with a score-based method [15] and with a noise-based method [16]), but we consider such hybrid methods as beyond the scope of this paper.

At the core of constraint-based approaches lie (in)dependence measures, to detect relevant dependencies, which are based here on an information theoretic approach. Since their introduction [17], information theoretic measures have become very popular due to their non-parametric nature, their robustness against strictly monotonic transformations, which makes them capable of handling nonlinear distortions in the system, and their good behavior in previous studies on causal discovery [18]. However, their application to temporal data raises several problems related to the fact that time series may have different sampling rates, be shifted in time, and have strong internal dependencies. Many studies have attempted to re-formalize mutual information for time series. Reference [19] considered each value of each time series as different random variables and proceeded by whitening the data, such that time-dependent data will be transformed into independent residuals through a parametric model. However, whitening the data can have severe consequences on causal relations. Reference [20] proposed a reformulation of mutual information, called the transfer entropy, which represents the information flow from one state to another and, thus, is asymmetric. Later, Ref. [13] generalized transfer entropy to handle conditioning. However, their approach, called causation entropy, can only handle causal relations with lags equal to 1. Closely related, the directed information [21,22] allows the computation of the mutual information between two instantaneous time series conditioned on the past of a third time series with a lag of 1. One of our goals in this paper is to extend these approaches to handle any lag equal to or greater than 0. Another related approach is the time-delayed mutual information proposed in [23], which aims at addressing the problem of non-uniform sampling rates. The computation of the time-delayed mutual information relates single points from a single time series (shifted in time), but does not consider potentially complex relations between time stamps in different time series, as we do through the use of window-based representations and compatible time lags. The measure we propose is more suited to discovering summary causal graphs as it can consider potentially complex relations between timestamps in different time series through the use of window-based representations and compatible time lags and is more general as it can consider different sampling rates.

## 3. Information Measures for Causal Discovery in Time Series

We present in this section a new mutual information measure that operates on a window-based representation of time series to assess whether time series are (conditionally) dependent or not. We then show how this measure is related to an entropy reduction principle that is a special case of the probability raising principle [24].

We first assume that all time series are aligned in time, with the same sampling rate, prior to showing how our development can be applied to time series with different sampling rates. Without loss of generality, time instants are assumed to be integers. Lastly, as performed in previous studies [20], we assume that all time series are first-order Markov self-causal (any time instant is caused by its previous instant within the same time series).

### 3.1. Causal Temporal Mutual Information

Let us consider *d* univariate time series $X^{1},\cdots ,X^{d}$ and their observations ${\left(X_{t}^{p}\right)}_{1\leq t\leq N_{p},1\leq p\leq d}$. Throughout this section, we will make use of the following Example 1, illustrated in Figure 1, to discuss the notions we introduce.

**Example** **1.***Let us consider the following two time series defined by, for all t,*$$\begin{array}{cc} X_{t}^{1} & =X_{t-1}^{1}+\xi_{t}^{1},\\ X_{t}^{2} & =X_{t-1}^{2}+X_{t-2}^{1}+X_{t-1}^{1}+\xi_{t}^{2}, \end{array}$$*with*$\left(\xi_{t}^{1},\xi_{t}^{2}\right)∼N\left(0,1\right)$.

One can see in Example 1 that, in order to capture the dependencies between the two time series, one needs to take into account a lag between them, as the true causal relations are not instantaneous. Several studies have recognized the importance of taking into account lags to measure (conditional) dependencies between time series; for example, in [7], a pointwise mutual information between time series with lags was used to assess whether they are dependent or not.

In addition to lags, Example 1 also reveals that a window-based representation may be necessary to fully capture the dependencies between the two time series. Indeed, as $X_{t-1}^{2}$ and $X_{t}^{2}$ are the effects of the same cause ($X_{t-2}^{1}$), it may be convenient to consider them together when assessing whether the time series are dependent or not. For example, defining (overlapping) windows of size two for $X^{2}$ and one for $X^{1}$ with a lag of 1 from $X^{1}$ to $X^{2}$, as in Figure 2, allows one to fully represent the causal dependencies between the two time series.

**Definition** **3.***Let*$\gamma_{\max}$*denote the maximum lag between two time series*$X^{p}$*and*$X^{q}$*, and let the maximum window size*$\lambda_{\max}=\gamma_{\max}+1$*. The window-based representation, of size*$0<\lambda_{pq}\leq \lambda_{\max}<N_{p}$*, of the time series*$X^{p}$*with respect to*$X^{q}$*, which will be denoted*$X^{\left(p;\lambda_{pq}\right)}$*, simply amounts to considering*$\left(N_{p}-\lambda_{pq}+1\right)$*windows:*$\left(X_{t}^{p},\cdots ,X_{t+\lambda_{pq}-1}^{p}\right),\;\;1\leq t\leq N_{p}-\lambda_{pq}+1$*. The window-based representation, of size*$0<\lambda_{qp}\leq \lambda_{\max}<N_{q}$*, of the time series*$X^{q}$*with respect to*$X^{p}$*is defined in the same way. A temporal lag*$\gamma_{pq}\in Z$ compatible *with*
$\lambda_{pq}$
*and*
$\lambda_{qp}$
*relates windows in*
$X^{\left(p;\lambda_{pq}\right)}$
*and*
$X^{\left(q;\lambda_{qp}\right)}$
*with starting time points separated by*
$\gamma_{pq}$*. We denote by*
$C^{\left(p,q\right)}$
*the set of window sizes and compatible temporal lags.*

Based on the above elements, we define the *causal temporal mutual information* between two time series $X^{p}$ and $X^{q}$ as the maximum of the standard mutual information over all possible compatible temporal lags and windows $C^{\left(p,q\right)}$, conditioned by the past of the two time series. Indeed, as we are interested in obtaining a summary causal graph, we do not have to consider all the potential dependencies between two time series (which would be necessary for inferring a window causal graph). Using the maximum over all possible associations is a way to summarize all temporal dependencies, which ensures that one does not miss a dependency between the two time series. Furthermore, conditioning on the past allows one to eliminate spurious dependencies in the form of auto-correlation, as in transfer entropy [20]. We follow this idea here and, as in transfer entropy, consider windows of size 1 and a temporal lag of 1 for conditioning on the past, which is in line with the first-order Markov self-causal assumption mentioned above.

**Definition** **4.***Consider two time series $X^{p}$ and $X^{q}$. We define the* causal temporal mutual information *between $X^{p}$ and $X^{q}$ as:*
(1)$$\begin{array}{cc} \mathrm{CTMI}(X^{p};X^{q})= & \max_{\left(\lambda_{pq},\lambda_{qp},\gamma_{pq}\right)\in C^{\left(p,q\right)}}I\left(X_{t}^{\left(p;\lambda_{pq}\right)};X_{t+\gamma_{pq}}^{\left(q;\lambda_{qp}\right)}|X_{t-1}^{\left(p;1\right)},X_{t+\gamma_{pq}-1}^{\left(q;1\right)}\right)\\ \overset{\Delta }{=} & I(X_{t}^{\left(p;{\bar{\lambda }}_{pq}\right)};X_{t+{\bar{\gamma }}_{pq}}^{\left(q;{\bar{\lambda }}_{qp}\right)}|X_{t-1}^{\left(p;1\right)};X_{t+{\bar{\gamma }}_{pq}-1}^{\left(q;1\right)}), \end{array}$$
*where I represents the mutual information. In case the maximum can be obtained with different values in $C^{\left(p,q\right)}$, we first set ${\bar{\gamma }}_{pq}$ to its largest possible value. We then set ${\bar{\lambda }}_{pq}$ to its smallest possible value and, finally, ${\bar{\lambda }}_{qp}$ to its smallest possible value. ${\bar{\gamma }}_{pq}$, ${\bar{\lambda }}_{pq}$, and ${\bar{\lambda }}_{qp}$, respectively, correspond to the optimal lag and optimal windows.*

In the context we have retained, in which dependencies are constant over time, CTMI satisfies the standard properties of mutual information, namely it is nonnegative, symmetric, and equals 0 *iff* time series are independent. Thus, two time series $X^{p}$ and $X^{q}$ such that $CTMI(X^{p};X^{q})>0$ are dependent. Setting ${\bar{\gamma }}_{pq}$ to its largest possible value allows one to get rid of instants that are not crucial in determining the mutual information between two time series. The choice for the window sizes, even though arbitrary on the choice of treating one window size before the other, is based on the same ground, as the mutual information defined above cannot decrease when one increases the size of the windows. Indeed: (2)$$\begin{array}{c} I(X_{t}^{\left(p;\lambda_{pq}\right)};X_{t+\gamma_{pq}}^{\left(q;\lambda_{qp}\right)}|X_{t-1}^{\left(p;1\right)},X_{t+\gamma_{pq}-1}^{\left(q;1\right)})\\ =I(\left(X_{t}^{\left(p;\lambda_{pq}-1\right)},X_{t+\lambda_{pq}-1}^{\left(p;1\right)}\right);X_{t+\gamma_{pq}}^{\left(q;\lambda_{qp}\right)}|X_{t-1}^{\left(p;1\right)},X_{t+\gamma_{pq}-1}^{\left(q;1\right)})\\ =I\left(X_{t}^{\left(p;\lambda_{pq}-1\right)};X_{t+\gamma_{pq}}^{\left(q;\lambda_{qp}\right)}|X_{t-1}^{\left(p;1\right)},X_{t+\gamma_{pq}-1}^{\left(q;1\right)}\right)+I\left(X_{t+\lambda_{pq}-1}^{\left(p;1\right)};X_{t+\gamma_{pq}}^{\left(q;\lambda_{qp}\right)}|X_{t-1}^{\left(p;1\right)},X_{t+\gamma_{pq}-1}^{\left(q;1\right)},X_{t}^{\left(p;\lambda_{pq}-1\right)}\right)\\ \geq I(X_{t}^{\left(p;\lambda_{pq}-1\right)};X_{t+\gamma_{pq}}^{\left(q;\lambda_{qp}\right)}|X_{t-1}^{\left(p;1\right)},X_{t+\gamma_{pq}-1}^{\left(q;1\right)}). \end{array}$$

The last inequality is due to the fact that mutual information is positive.

**Example** **2.***Consider the structure described in Example 1, and assume that $\lambda_{\max}=3$. First, we have, for the standard mutual information,*$$I(X_{t}^{\left(1;1\right)};X_{t}^{\left(2;1\right)}|X_{t-1}^{\left(1;1\right)},X_{t-1}^{\left(2;1\right)})=0.$$*We also have that any $\gamma_{12}<0$ has zero mutual information because conditioning on the past of $X_{t}^{\left(1;1\right)};X_{t}^{\left(2;1\right)}$ (namely $X_{t-1}^{\left(1;1\right)};X_{t-1}^{\left(2;1\right)}$) is closing all paths from $X_{t-i}^{\left(2;1\right)}$ to $X_{t}^{\left(1;1\right)}$ for all i>0. For $\gamma_{12}>0$, starting by $\gamma_{12}=1$,*$$I(X_{t}^{\left(1;1\right)};X_{t+1}^{\left(2;1\right)}|X_{t-1}^{\left(1;1\right)},X_{t}^{\left(2;1\right)})>0.$$*This is similar for $\gamma_{12}>2$. Now, any increase of $\lambda_{21}$ alone or of $\lambda_{12}$ and $\lambda_{21}$ will generate an increase in the mutual information as long as the difference between the last time point of the window of $X^{1}$ (the cause) and the last time point of the window of $X^{2}$ is less than or equal to $\gamma_{12}$ as*$$\begin{array}{c} I\left(X_{t}^{\left(1;\lambda_{12}-1\right)};X_{t+\gamma_{12}}^{\left(2;\lambda_{12}-1-\gamma_{12}\right)}|X_{t-1}^{\left(1;1\right)},X_{t+\gamma_{12}-1}^{\left(2;1\right)}\right)=I\left(X_{t}^{\left(1;\lambda_{12}\right)};X_{t+\gamma_{12}}^{\left(2;\lambda_{12}-1-\gamma_{12}\right)}|X_{t-1}^{\left(1;1\right)},X_{t+\gamma_{12}-1}^{\left(2;1\right)}\right) \end{array}$$*because*$$I(X_{t+\lambda_{12}-1}^{\left(1;1\right)};X_{t+\gamma_{12}}^{\left(2;\lambda_{12}-1-\gamma_{12}\right)}|X_{t-1}^{\left(1;1\right)},X_{t+\gamma_{12}-1}^{\left(2;1\right)},X_{t}^{\left(1;\lambda_{12}-1\right)})=0,$$*where $\gamma_{12}\geq 1$ (*1 *is the minimal lag that generates a correlation that cannot be removed by conditioning on the past of $X^{1}$ and $X^{2}$). For $\lambda_{\max}=3$, the optimal window size ${\bar{\lambda }}_{12}$ is equal to 2 as $X^{1}$ has no other cause than itself; ${\bar{\lambda }}_{21}$ is equal to 2 as $X^{1}$ causes only (except itself) $X_{t+1}^{2}$ and $X_{t}^{2}$. Furthermore, ${\bar{\gamma }}_{12}=1$ and*$$\begin{array}{cc} \mathrm{CTMI}(X^{1};X^{2}) & =I(\left(X_{t}^{1},X_{t+1}^{1}\right);\left(X_{t+1}^{2},X_{t+2}^{2}\right)|X_{t-1}^{1},X_{t}^{2})\\ \; & =I\left(X_{t}^{1};\left(X_{t+1}^{2},X_{t+2}^{2}\right)|X_{t-1}^{1},X_{t}^{2}\right)+I\left(X_{t+1}^{1};\left(X_{t+1}^{2},X_{t+2}^{2}\right)|X_{t-1}^{1},X_{t}^{2},X_{t}^{1}\right)\\ \; & =I\left(X_{t}^{1};X_{t+1}^{2}|X_{t-1}^{1},X_{t}^{2}\right)+I\left(X_{t}^{1};X_{t+2}^{2}|X_{t-1}^{1},X_{t}^{2},X_{t+1}^{2}\right)\\ \; & \;+I\left(X_{t+1}^{1};X_{t+1}^{2}|X_{t-1}^{1},X_{t}^{2},X_{t}^{1}\right)+I\left(X_{t+1}^{1};X_{t+2}^{2}|X_{t-1}^{1},X_{t}^{2},X_{t}^{1},X_{t+1}^{2}\right)\\ \; & =2I\left(X_{t}^{1};X_{t+1}^{2}|X_{t-1}^{1},X_{t}^{2}\right)+I\left(X_{t}^{1};X_{t+2}^{2}|X_{t-1}^{1},X_{t}^{2},X_{t+1}^{2}\right)=3\log\left(3\right)/4. \end{array}$$

### 3.2. Entropy Reduction Principle

Interestingly, CTMI can be related to a version of the *probability raising principle* [24], which states that a cause, here a time series, raises the probability of any of its effects, here another time series, even when the past of the two time series is taken into account, meaning that the relation between the two time series is not negligible compared to the internal dependencies of the time series. In this context, the following definition generalizes to window-based representations of time series the standard definition of prima facie causes for discrete variables.

**Definition** **5****(Prima facie cause for window-based time series).***Let $X^{p}$ and $X^{q}$ be two time series with window sizes $\lambda_{pq}$ and $\lambda_{qp}$, and let $P_{t,t^{\prime }}=\left(X_{t-1}^{\left(p;1\right)},X_{t^{\prime }-1}^{\left(q;1\right)}\right)$ represent the past of $X^{p}$ and $X^{q}$ for any two instants $\left(t,t^{\prime }\right)$. We say that $X^{p}$ is a* prima facie cause *of $X^{q}$ with delay $\gamma_{pq}>0$ iff there exist Borel sets $B_{p}$, $B_{q}$, and $B_{P}$ such that one has:*
$P\left(X_{t+\gamma_{pq}}^{\left(q;\lambda_{qp}\right)}\in B_{q}|X_{t}^{\left(p;\lambda_{pq}\right)}\in B_{p},P_{t,t+\gamma_{pq}}\in B_{P}\right)>P\left(X_{t+\gamma_{pq}}^{\left(q;\lambda_{qp}\right)}\in B_{q}|P_{t,t+\gamma_{pq}}\in B_{P}\right).$

We now introduce a slightly different principle based on the causal temporal mutual information, which we refer to as the *entropy reduction principle*.

**Definition** **6****(Entropic prima facie cause).***Using the same notations as in Definition 5, we say that $X^{p}$ is an* entropic prima facie cause *of $X^{q}$ with delay $\gamma_{pq}>0$ iff $I(X_{t}^{\left(p;\lambda_{pq}\right)};X_{t+\gamma_{pq}}^{\left(q;\lambda_{qp}\right)}|P_{t,t+\gamma_{pq}})>0$.*

Note that considering that the above mutual information is positive is equivalent to considering that the entropy of $X^{q}$ when conditioned on the past reduces when one further conditions on $X^{p}$. One has the following relation between the entropy reduction and the probability raising principles.

**Property** **1.***With the same notations, if $X^{p}$ is an* entropic prima facie *cause of $X^{q}$ with delay $\gamma_{pq}>0$, then $X^{p}$ is a prima facie cause of $X^{q}$ with delay $\gamma_{pq}>0$. Furthermore, if $\mathrm{CTMI}(X^{p};X^{q})>0$ with ${\bar{\gamma }}_{pq}>0$, then $X^{p}$ is an* entropic prima facie *cause of $X^{q}$ with delay ${\bar{\gamma }}_{pq}$.*

**Proof.** Let us assume that $X^{p}$ is not a prima facie cause of $X^{q}$ for the delay $\gamma_{pq}$. Then, for all Borel sets $B_{p}$, $B_{q}$, and $B_{P}$, one has $P\left(X_{t+\gamma_{pq}}^{\left(q;\lambda_{qp}\right)}\in B_{q}|X_{t}^{\left(p;\lambda_{pq}\right)}\in B_{p},P_{t,t+\gamma_{pq}}\in B_{P}\right)\leq P\left(X_{t+\gamma_{pq}}^{\left(q;\lambda_{qp}\right)}\in B_{q}|P_{t,t+\gamma_{pq}}\in B_{P}\right)$. This translates, in terms of density functions denoted *f*, as:
$$\forall \left(x_{t}^{p},x_{t+\gamma_{pq}}^{q},p_{t,t+\gamma_{pq}}\right),\;\;f\left(x_{t+\gamma_{pq}}^{q}|x_{t}^{p},p_{t,t+\gamma_{pq}}\right)\leq f\left(x_{t+\gamma_{pq}}^{q}|p_{t,t+\gamma_{pq}}\right),$$
which implies that $H(X_{t+\gamma_{pq}}^{\left(q;\lambda_{qp}\right)}\in B_{q}|X_{t}^{\left(p;\lambda_{pq}\right)}\in B_{p},P_{t,t+\gamma_{pq}}\in B_{P})$ is greater than $H(X_{t+\gamma_{pq}}^{\left(q;\lambda_{qp}\right)}\in B_{q}|P_{t,t+\gamma_{pq}}\in B_{P})$, so that $X^{p}$ is not an *entropic prima facie* cause of $X^{p}$ with delay $\gamma_{pq}$. By contraposition, we conclude the proof of the first statement. The second statement directly derives from the definition of CTMI.  □

It is also interesting to note that, given two time series $X^{p}$ and $X^{q}$ such that $X^{p}$ causes $X^{q}$ with $\gamma_{pq}=0$, CTMI does not necessarily increase symmetrically with respect to the increase of $\lambda_{pq}$ and $\lambda_{qp}$. For an illustration on a simple model, see Figure 3. In Figure 3a, the mutual information conditioned on the past between $X^{p}$ and $X^{q}$ with $\gamma_{pq}=0$ is positive since $X^{p}$ causes $X^{q}$ instantaneously. In Figure 3b the same mutual information does not increase when increasing only the window size of the cause. However, the mutual information increases when increasing only the window size of the effect, as in Figure 3c, or when increasing simultaneously the window sizes of the effect and the cause, as in Figure 3d.

### 3.3. Conditional Causal Temporal Mutual Information

We now extend the causal temporal mutual information by conditioning on a set of variables. In a causal discovery setting, conditioning is used to assess whether two dependent time series can be made independent by conditioning on connected time series, i.e., time series that are dependent with at least one of the two times series under consideration. Figure 4 illustrates the case where the dependence between $X^{1}$ and $X^{2}$ is due to spurious correlations originating from common causes. Conditioning on these common causes should lead to the conditional independence of the two time series. Of course, the conditional variables cannot succeed simultaneously in time the two time series under consideration. This leads us to the following definition of the conditional causal temporal mutual information.

**Definition** **7.***The* conditional causal temporal mutual information *between two time series $X^{p}$ and $X^{q}$ such that ${\bar{\gamma }}_{pq}\geq 0$, conditioned on a set $X^{R}=\left\{X^{r_{1}},\cdots ,X^{r_{K}}\right\}$, is given by:*
(3)$$\begin{array}{c} \mathrm{CTMI}\left(X^{p};X^{q}|X^{R}\right)=I\left(X_{t}^{\left(p;{\bar{\lambda }}_{pq}\right)};X_{t+{\bar{\gamma }}_{pq}}^{\left(q;{\bar{\lambda }}_{qp}\right)}|{\left(X_{t-{\bar{\Gamma }}_{k}}^{\left(r_{k};{\bar{\lambda }}_{k}\right)}\right)}_{1\leq k\leq K},X_{t-1}^{\left(p;1\right)},X_{t+{\bar{\gamma }}_{pq}-1}^{\left(q;1\right)}\right). \end{array}$$
*In case the minimum can be obtained with different values, we first set ${\bar{\Gamma }}_{k}$ to its largest possible value. We then set ${\bar{\lambda }}_{k}$ to its smallest possible value. $\left({\bar{\Gamma }}_{1},\ldots ,{\bar{\Gamma }}_{K}\right)$ and $\left({\bar{\lambda }}_{1},\cdots ,{\bar{\lambda }}_{K}\right)$ correspond to the optimal conditional lags and window sizes, which minimize, for $\Gamma_{1},\ldots ,\Gamma_{K}\geq -{\bar{\gamma }}_{pq}$:*
$$\begin{array}{c} I\left(X_{t}^{\left(p;{\bar{\lambda }}_{pq}\right)};X_{t+{\bar{\gamma }}_{pq}}^{\left(q;{\bar{\lambda }}_{qp}\right)}|{\left(X_{t-\Gamma_{k}}^{\left(r_{k};\lambda_{k}\right)}\right)}_{1\leq k\leq K},X_{t-1}^{\left(p;1\right)},X_{t+{\bar{\gamma }}_{pq}-1}^{\left(q;1\right)}\right). \end{array}$$

By considering the minimum over compatible lags and window sizes, one guarantees that, if there exist conditioning variables that make the two time series independent, they will be found. Note that the case in which ${\bar{\gamma }}_{qp}<0$ corresponds to $\mathrm{CTMI}(X^{p};X^{q}|X^{R})$, where ${\bar{\gamma }}_{pq}>0$.

Figure 4 illustrates the above on two different examples. On the left, $X_{t-1}^{1}$ is correlated to $X_{t}^{2}$ as $X_{t-2}^{3}$ is a common cause with a lag of 1 for $X^{1}$ and a lag of 2 for $X^{2}$. Conditioning on $X_{t-2}^{3}$ removes the dependency between $X^{1}$ and $X^{2}$. Note that all time series have here a window of size 1. On the right, $X^{3}$ and $X^{4}$ are common causes of $X^{1}$ and $X^{2}$: $X^{3}$ causes $X^{1}$ and $X^{2}$ with a temporal lag of 1, which renders $X^{1}$ and $X^{2}$ correlated at the same time point, while $X^{4}$ causes $X^{1}$ and $X^{2}$ with a temporal lag of 1 and 2, respectively, which renders $X^{1}$ and $X^{2}$ correlated at lagged time points. The overall correlation between $X^{1}$ and $X^{2}$ is captured by considering a window size of 2 in $X^{2}$. All other time series have a window size of 1. By conditioning on both $X^{3}$ and $X^{4}$, $X^{p}$ and $X^{q}$ become independent, assuming we also condition on the past of $X^{1}$ and $X^{2}$ to remove the autocorrelation.

### 3.4. Estimation and Testing

In practice, the success of the CTMI approach (and in fact, any entropy-based approaches) depends crucially on the reliable estimation of the relevant entropies in question from data. This leads to two practical challenges. The first one is based on the fact that entropies must be estimated from finite-time series data. The second is that to detect independence, we need a statistical test to check if the temporal causation entropy is equal to zero.

Here, we rely on the *k*-nearest neighbor method [25,26] for the estimation of CTMI. The distance between two windows considered here is the supremum distance, i.e., the maximum of the absolute difference between any two values in the two windows.
$$\begin{array}{cc} d\left({\left(X_{t}^{\left(p;{\bar{\lambda }}_{pq}\right)},X_{t+{\bar{\gamma }}_{pq}}^{\left(q;{\bar{\lambda }}_{pq}\right)}\right)}_{i},{\left(X_{t}^{\left(p;{\bar{\lambda }}_{pq}\right)},X_{t+{\bar{\gamma }}_{pq}}^{\left(q;{\bar{\lambda }}_{pq}\right)}\right)}_{j}\right)=\max_{\begin{array}{c} 0\leq \ell <\lambda_{p},\;0\leq \ell^{\prime }<\lambda_{q} \end{array}}( & |{\left(X_{t}^{\left(p;{\bar{\lambda }}_{pq}\right)}\right)}_{i+\ell }-{\left(X_{t}^{\left(p;{\bar{\lambda }}_{pq}\right)}\right)}_{j+\ell^{\prime }}|,\\ & |{\left(X_{t}^{\left(q;{\bar{\lambda }}_{qp}\right)}\right)}_{i+\ell^{\prime }}-{\left(X_{t}^{\left(q;{\bar{\lambda }}_{qp}\right)}\right)}_{j+\ell^{\prime }}\left|\right). \end{array}$$
In the case of the causal temporal mutual information, we denote by $\epsilon_{ik}/2$ the distance from
$$\left(X_{t}^{\left(p;{\bar{\lambda }}_{pq}\right)},X_{t+{\bar{\gamma }}_{pq}}^{\left(q;{\bar{\lambda }}_{pq}\right)},X_{t-1}^{p},X_{t+{\bar{\gamma }}_{pq}-1}^{q}\right)$$
to its *k*-th neighbor, $n_{i}^{1,3}$, $n_{i}^{2,3}$, and $n_{i}^{3}$ being the numbers of points with a distance strictly smaller than $\epsilon_{ik}/2$ in the subspace
$$\left(X_{t}^{\left(p;{\bar{\lambda }}_{pq}\right)},X_{t-1}^{p},X_{t+{\bar{\gamma }}_{pq}-1}^{q}\right),\left(X_{t+{\bar{\gamma }}_{pq}}^{\left(q;{\bar{\lambda }}_{pq}\right)},X_{t-1}^{p},X_{t+{\bar{\gamma }}_{pq}-1}^{q}\right),\;\mathrm{and}\;\left(X_{t-1}^{p},X_{t+{\bar{\gamma }}_{pq}-1}^{q}\right)$$
respectively, and $n_{\gamma_{pq},\gamma_{qp}}$ the number of observations. The estimate of the causal temporal mutual information is then given by:$$\begin{array}{c} \hat{\mathrm{CTMI}}\left(X^{p};X^{q}\right)=\psi \left(k\right)+\frac{1}{n_{\gamma_{pq},\gamma_{qp}}}\sum_{i=1}^{n_{\gamma_{pq},\gamma_{qp}}}\psi \left(n_{i}^{3}\right)-\psi \left(n_{i}^{1,3}\right)-\psi \left(n_{i}^{2,3}\right) \end{array}$$
where ψ denotes the digamma function.

Similarly, for the estimation of the conditional causal temporal mutual information, we denote by $\epsilon_{ik}/2$ the distance from
$$\left(X_{t}^{\left(p;{\bar{\lambda }}_{pq}\right)},X_{t+{\bar{\gamma }}_{pq}}^{\left(q;{\bar{\lambda }}_{pq}\right)},X_{t-1}^{p},X_{t+{\bar{\gamma }}_{pq}-1}^{q},{\left(X_{t-{\bar{\Gamma }}_{k}}^{\left(r_{k};{\bar{\lambda }}_{k}\right)}\right)}_{1\leq k\leq K}\right)$$
to its *k*-th neighbor, $n_{i}^{1,3}$, $n_{i}^{2,3}$, and $n_{i}^{3}$ being the numbers of points with a distance strictly smaller than $\epsilon_{ik}/2$ in the subspace
$$\begin{array}{cc} \left(X_{t}^{\left(p;{\bar{\lambda }}_{pq}\right)},X_{t-1}^{p},X_{t+{\bar{\gamma }}_{pq}-1}^{q},{\left(X_{t-{\bar{\Gamma }}_{k}}^{\left(r_{k};{\bar{\lambda }}_{k}\right)}\right)}_{1\leq k\leq K}\right) & ,(X_{t+{\bar{\gamma }}_{pq}}^{\left(q;{\bar{\lambda }}_{pq}\right)},X_{t-1}^{p},X_{t+{\bar{\gamma }}_{pq}-1}^{q},{\left(X_{t-{\bar{\Gamma }}_{k}}^{\left(r_{k};{\bar{\lambda }}_{k}\right)}\right)}_{1\leq k\leq K}),\\ \; & \;\mathrm{and}\;(X_{t-1}^{p},X_{t+{\bar{\gamma }}_{pq}-1}^{q},{\left(X_{t-{\bar{\Gamma }}_{k}}^{\left(r_{k};{\bar{\lambda }}_{k}\right)}\right)}_{1\leq k\leq K}) \end{array}$$
respectively, and $n_{\gamma_{rp},\gamma_{rq}}$ the number of observations. The estimate of the conditional causal temporal mutual information is then given by:
$$\begin{array}{cc} \; & \hat{CTMI}\left(X^{p};X^{q}|X^{R}\right)=\psi \left(k\right)+\frac{1}{n_{\gamma_{rp},\gamma_{rq}}}\sum_{i=1}^{n_{\gamma_{rp},\gamma_{rq}}}\psi \left(n_{i}^{3}\right)-\psi \left(n_{i}^{1,3}\right)-\psi \left(n_{i}^{2,3}\right) \end{array}$$
where ψ denotes the digamma function.

To detect independencies through CTMI, we rely on the following permutation test:

**Definition** **8**(Permutation test of CTMI)**.**
*Given $X^{p}$, $X^{q}$, and $X^{R}$, the p-value associated with the permutation test of CTMI is given by:*
(4)$$p=\frac{1}{B}\sum_{b=1}^{B}1_{\hat{CTMI}\left({\left(X^{p}\right)}_{b};X^{q}|X^{R}\right)\geq \hat{CTMI}\left(X^{p};X^{q}|X^{R}\right)},$$
*where ${\left(X^{p}\right)}_{b}$ is a permuted version of $X^{p}$ and follows the local permutation scheme presented in [27].*

The advantage of the scheme presented in [27] is that it preserves marginals by drawing as much as possible without replacement and it performs local permutation, which ensures that by permuting $X^{p}$, the dependence between $X^{p}$ and $X^{r}$ is not destroyed.

Note that Definition 8 is applicable to the causal temporal mutual information (when R is empty) and to the conditional causal temporal mutual information.

### 3.5. Extension to Time Series with Different Sampling Rates

The above development readily applies to time series with different sampling rates, as one can define window-based representations of the two time series, as well as a sequence of joint observations.

Indeed, as one can note, Definition 3 does not rely on the fact that the time series have the same sampling rates. Figure 5 displays two time series $X^{p}$ and $X^{q}$ with different sampling rates, where, while $\lambda_{pq}=2$ and $\lambda_{qp}=3$, the time spanned by each window is the same. The joint sequence of observations, relating pairs of windows from $X^{p}$ and $X^{q}$ in the form $S=\{\left(X_{1_{p}}^{\left(p;\lambda_{pq}\right)},X_{1_{q}}^{\left(q;\lambda_{qp}\right)}\right),\cdots ,\left(X_{n_{p}}^{\left(p;\lambda_{pq}\right)},X_{n_{q}}^{\left(q;\lambda_{qp}\right)}\right)\}$, should, however, be such that, for all indices *i* of the sequence, one has: $s\left(X_{i_{q}}^{\left(q;\lambda_{qp}\right)}\right)=s\left(X_{i_{p}}^{\left(p;\lambda_{pq}\right)}\right)+\gamma_{pq}$, where s(X) represents the starting time of the window *X*, and $\gamma_{pq}$ is constant over time. This is not the case for the first example, but is true for the second one, which is a relevant sequence of observations.

If the two time series are sufficiently long, there always exists a correct sequence of joint observations. Indeed, if the window sizes $\lambda_{pq}$ and $\lambda_{qp}$ are known, let $\gamma_{pq}=s\left(X_{1}^{\left(q;\lambda_{qp}\right)}\right)-s\left(X_{1}^{\left(p;\lambda_{pq}\right)}\right)$. Furthermore, let $N_{p}$ and $N_{q}$ denote the number of observations per time unit (the time unit corresponds to the largest (integer) time interval according to the sampling rates of the different time series. For example, if a time series has a sampling rate of 10 per second and another a sampling rate of 3 per 10 min, then the time unit is equal to 10 min). Then, $\lambda_{pq}$, $\lambda_{qp}$, and $\gamma_{pq}$ are compatible through the set of joint observations *S* with $s\left(X_{i_{p}}^{\left(p;\lambda_{pq}\right)}\right)=s\left(X_{1}^{\left(p;\lambda_{pq}\right)}\right)+\left(i_{p}-1\right)LCM\left(N_{p},N_{q}\right)$ and $s\left(X_{i_{q}}^{\left(q;\lambda_{qp}\right)}\right)=s\left(X_{1}^{\left(q;\lambda_{qp}\right)}\right)+\left(i_{q}-1\right)LCM\left(N_{p},N_{q}\right),$ such that LCM is the lowest common multiple.

Note that this methodology for handling different sampling rates is not unique to our proposal and can be used as soon as lags and windows are defined.

## 4. Causal Discovery Based on Causal Temporal Mutual Information

We present in this section two new methods for causal discovery in time series based on the causal temporal mutual information introduced above to construct the skeleton of the causal graph. The first method assumes causal sufficiency (there exists no hidden common cause of two observable time series), while the second method is an extension of the first for the case where causal sufficiency is not satisfied. In both methods, the skeleton is oriented on the basis of the entropy reduction principle in addition to classical constraint-based orientation rules (the rules used in the PC algorithm or the rules used in the FCI algorithm). Our methods assume both the causal Markov condition and faithfulness of the data distribution, which are classical assumptions for causal discovery within constraint-based methods in addition to acyclicity in the summary causal graph.

### 4.1. Without Hidden Common Causes

We present here our main method, which assumes causal sufficiency. We start by describing how we infer the skeleton of the summary graph, then we show how we orient edges using classical PC-rules in addition to rules based on the entropy reduction principle.

#### 4.1.1. Skeleton Construction

We follow the same steps as the ones of the PC algorithm [9], which assumes that all variables are observed. It aims at building causal graphs by orienting a skeleton obtained, from a complete graph, by removing edges connecting independent variables. The summary causal graphs considered are directed acyclic graphs (DAGs) in which self-loops are allowed to represent temporal dependencies within a time series.

Starting with a complete graph that relates all time series, the first step consists of computing CTMI for all pairs of time series and removing edges if the two time series are considered independent. Once this is done, one checks, for the remaining edges, whether the two time series are conditionally independent (the edge is removed) or not (the edge is kept). Starting from a single time series connected to $X^{p}$ or $X^{q}$, the set of conditioning time series is gradually increased until either the edge between $X^{p}$ and $X^{q}$ is removed or all time series connected to $X^{p}$ and $X^{q}$ have been considered. We will denote by Sepset(p,q) the separation set of $X^{p}$ and $X^{q}$, which corresponds to the smallest set of time series connected to $X^{p}$ and $X^{q}$ such that $X^{p}$ and $X^{q}$ are conditionally independent given this set. Note that we make use of the same strategy as the one used in PC-stable [28], which consists of sorting time series according to their CTMI scores and, when an independence is detected, removing all other occurrences of the time series. This leads to an order-independent procedure. The following theorem states that the skeleton obtained by the above procedure is the true one.

**Theorem** **1.**
*Let G=(V,E) be a summary causal graph, and assume that we are given perfect conditional independence information about all pairs of variables $\left(X^{p},X^{q}\right)$ in V given subsets $S\subseteq V\setminus \{X^{p},X^{q}\}$. Then, the skeleton previously constructed is the skeleton of G.*


**Proof.** Let us consider two time series $X^{p}$ and $X^{q}$. If they are independent given $X^{R}$, then $\mathrm{CTMI}(X^{p};X^{q}|X^{R})=0$, as otherwise, the conditional mutual information between $X^{p}$ and $X^{q}$ would be non-null and the two time series would not be conditionally independent as we are given perfect information. By the causal Markov and faithfulness conditions, there is no edge in this case between $X^{p}$ and $X^{q}$ in the corresponding skeleton, as in the true one. Conversely, if $\mathrm{CTMI}(X^{p};X^{q}|X^{R})=0$ for any $X^{R}$, then the two time series cannot be dependent conditioned on $X^{R}$. Indeed, if this were the case, as we are given perfect conditional information, there would exist a lag γ and two window sizes $\lambda_{pq}$ and $\lambda_{pq}$ such that $I(X_{t}^{\left(p;\lambda_{pq}\right)};X_{t+\gamma }^{\left(q;\lambda_{qp}\right)}|X^{R})>0$ with $0<\lambda_{pq},\lambda_{qp}\leq \gamma$. In this case, the two windows of size $\lambda_{max}$ centered on time point *t* in both $X^{p}$ and $X^{q}$ contain the windows of sizes $\lambda_{pq}$ and $\lambda_{pq}$ separated by a lag γ in $X^{p}$ and $X^{q}$ as $\lambda_{max}=2\gamma_{max}+1$. Thus, $\mathrm{CTMI}(X^{p};X^{q}|X^{R})$ would be positive as this quantity cannot be less than $I(X_{t}^{\left(p;\lambda_{pq}\right)};X_{t+\gamma }^{\left(q;\lambda_{qp}\right)}|X^{R})$, which leads to a contradiction. Finally, as we tested all necessary conditioning sets in the construction of the skeleton, we have the guarantee of removing all unnecessary edges.  □

#### 4.1.2. Orientation

Once the skeleton has been constructed, one tries to orient as many edges as possible using the standard PC-rules [9], which yields a completed partially directed acyclic graph (CPDAG).

**PC-rule** **0**(Origin of causality)**.**
*In an unshielded triple $X^{p}-X^{r}-X^{q}$, if $X^{r}\notin \mathrm{sepset}\left(p,q\right)$, then $X^{r}$ is an unshielded collider: $X^{p}\to X^{r}\leftarrow X^{q}$.*

**PC-rule** **1**(Propagation of causality)**.**
*In an unshielded triple $X^{p}\to X^{r}-X^{q}$, if $X^{r}\in \mathrm{sepset}\left(p,q\right)$, then orient the unshielded triple as $X^{p}\to X^{r}\to X^{q}$.*

**PC-rule** **2.**
*If there exist a direct path from $X^{p}$ to $X^{q}$ and an edge between $X^{p}$ and $X^{q}$, then orient $X^{p}\to X^{q}$.*


**PC-rule** **3.**
*If there exists an unshielded triple $X^{p}\to X^{r}\leftarrow X^{q}$ and an unshielded triple $X^{p}-X^{s}-X^{q}$, then orient $X^{s}\to X^{r}$.*


As we are using here the standard PC-rules, we have the following theorem, the proof of which directly derives from the results on PC [9].

**Theorem** **2**(Theorem 5.1 of [9])**.**
*Let the distribution of V be faithful to a DAG G=(V,E), and assume that we are given perfect conditional independence information about all pairs of variables $\left(X^{p},X^{q}\right)$ in V given subsets $X^{R}\subseteq V\setminus \left\{X^{p},X^{q}\right\}$. Then, the skeleton constructed previously followed by PC-rules 0, 1, 2, and 3 represents the CPDAG of G.*

In addition to the PC orientation rules, we introduce two new rules, which are based on the notion of *possible spurious correlations* and the mutual information we have introduced. The notion of possible spurious correlations captures the fact that two variables may be correlated through relations that do not only correspond to direct causal relations between them. It is formalized as follows:

**Definition** **9**(Possible spurious correlations)**.**
*We say that two nodes $X^{p}-X^{q}$ have possible spurious correlations if there exists a path between them that neither contains the edge $X^{p}-X^{q}$ nor any collider.*

Interestingly, when two connected variables do not have possible spurious correlations, one can conclude about their orientation using CTMI.

**Property** **2.**
*Let us assume that we are given perfect conditional independence information about all pairs of variables $\left(X^{p},X^{q}\right)$ in V given subsets $S\subseteq V\backslash \{X^{p},X^{q}\}$. Then, every non-oriented edge in the CPDAG obtained by the above procedure corresponds to a prima facie cause and by, causal sufficiency, to a true causal relation between the related time series. Furthermore, the orientation of an unoriented edge between two nodes $X^{p}$ and $X^{q}$ that do not have possible spurious correlations is given by the “direction” of the optimal lag in $\mathrm{CTMI}(X^{p},X^{q})$, assuming that the maximal window size is larger than the longest lag $\gamma_{max}$ between causes and effects.*


**Proof.** The first part of Property 2 directly derives from Property 1. As we assume that we are given perfect conditional information, the skeleton is the true one from Theorem 1. Thus, if two variables do not have possible spurious correlations, the only correlations observed between them correspond to a causal relation. We now need to prove that the optimal lag can be used to orient edges between any pair of variables $X^{p}$ and $X^{q}$.Without loss of generality, let us assume that $X^{p}$ causes $X_{t}^{q}$, for any time *t*, via the *K* time instants $\left\{t-\gamma ,t-\gamma_{1},\cdots ,t-\gamma_{K-1}\right\}$ with $0<\gamma_{K-1}<\cdots <\gamma_{1}<\gamma$. First, let us consider a window of size 1 in $X^{q}$ and a window of arbitrary size λ in $X^{p}$ with a lag $\gamma_{pq}$ set to $\gamma^{\prime }\geq 0$. As $\gamma^{\prime }\geq 0$, there is no cause of $X_{t}^{q}$ in the window $X_{t+\gamma^{\prime }}^{\left(p;\lambda \right)}$. Furthermore, the only observed correlations between $X^{p}$ and $X^{q}$ correspond to causal relations. We thus have:
$$I(X_{t}^{\left(q;1\right)};X_{t+\gamma^{\prime }}^{\left(p;\lambda \right)}|X_{t-1}^{\left(q;1\right)},X_{t+\gamma^{\prime }-1}^{\left(p;1\right)})=0,$$
as $X_{t}^{q}$ and all variables in $X_{t+\gamma^{\prime }}^{\left(p;\lambda \right)}$ are independent of each other. One the contrary, for the same window size in $X^{p}$ and a lag $\gamma_{pq}$ set to −γ with γ>0, one has:
$$I\left(X_{t}^{\left(q;1\right)};X_{t-\gamma }^{\left(p;\gamma \right)}|X_{t-1}^{\left(q;1\right)},X_{t-\gamma -1}^{\left(p;1\right)}\right)\geq I\left(X_{t}^{\left(q;1\right)};X_{t-\gamma }^{\left(p;1\right)}|X_{t-1}^{\left(q;1\right)},X_{t-\gamma -1}^{\left(p;1\right)}\right)>0.$$
The first inequality derives from Inequality Equation 2. The second inequality is due to the fact that $X_{t-\gamma }^{p}$ is a true cause of $X_{t}^{q}$ and the fact that we are given perfect information. Thus, when considering a window of size 1 for $X^{q}$, the optimal lag given by CTMI will necessarily go from $X^{p}$ to $X^{q}$, which corresponds to the correct orientation.We now consider the case where we have a window of arbitrary size $\lambda^{\prime }$ in $X^{q}$. Let us further consider a window of arbitrary size λ in $X^{p}$ with a lag $\gamma_{pq}$ set to $\gamma^{\prime }\geq 0$. If $\lambda^{\prime }<\gamma^{\prime }+\gamma_{K-1}$, there is no causal relations between the elements in $X_{t}^{\left(q;\lambda^{\prime }\right)}$ and the elements in $X_{t+\gamma^{\prime }}^{\left(p;\lambda \right)}$ and the mutual information between these two windows is 0. Otherwise, one can decompose this mutual information as:
$$\begin{array}{cc} \; & I(X_{t}^{\left(q;\lambda^{\prime }\right)};X_{t+\gamma^{\prime }}^{\left(p;\lambda \right)}|X_{t-1}^{\left(q;1\right)},X_{t+\gamma^{\prime }-1}^{\left(p;1\right)})\\ \; & =I\left(X_{t}^{\left(q;\gamma^{\prime }+\gamma_{K-1}\right)};X_{t+\gamma^{\prime }}^{\left(p;\lambda \right)}|X_{t-1}^{\left(q;1\right)},X_{t+\gamma^{\prime }-1}^{\left(p;1\right)}\right)+I\left(X_{t+\gamma^{\prime }+\gamma_{K-1}}^{\left(q;\lambda^{\prime }-\gamma^{\prime }-\gamma_{K-1}\right)};X_{t+\gamma^{\prime }}^{\left(p;\lambda \right)}|X_{t+\gamma^{\prime }+\gamma_{K-1}-1}^{\left(q;1\right)},X_{t+\gamma^{\prime }-1}^{\left(p;1\right)}\right), \end{array}$$
as the conditioning on $X_{t}^{\left(q;\gamma^{\prime }+\gamma_{K-1}\right)}$ and $X_{t-1}^{\left(q;1\right)}$ amounts to conditioning on the instant $X_{t+\gamma^{\prime }+\gamma_{K-1}-1}^{\left(q;1\right)}$ due to the first-order Markov self-causal assumption.As there are no causal relations between the elements in $X_{t}^{\left(q;\gamma^{\prime }+\gamma_{K-1}\right)}$ and the elements in $X_{t+\gamma^{\prime }}^{\left(p;\lambda \right)}$, the first term on the right-hand side is 0. Using a similar decomposition in order to exclude elements at the end of $X_{t+\gamma^{\prime }}^{\left(p;\lambda \right)}$, which do not cause any element in $X_{t}^{\left(q;\lambda^{\prime }\right)}$, one obtains:
$$\begin{array}{cc} \; & I(X_{t}^{\left(q;\lambda^{\prime }\right)};X_{t+\gamma^{\prime }}^{\left(p;\lambda \right)}|X_{t-1}^{\left(q;1\right)},X_{t+\gamma^{\prime }-1}^{\left(p;1\right)})\\ \; & =I(X_{t+\gamma^{\prime }+\gamma_{K-1}}^{\left(q;\lambda^{\prime }-\gamma^{\prime }-\gamma_{K-1}\right)};X_{t+\gamma^{\prime }}^{\left(p;\min\left(\lambda ,\lambda^{\prime }-\gamma_{K-1}-\gamma^{\prime }\right)\right)}|X_{t+\gamma^{\prime }+\gamma_{K-1}-1}^{\left(q;1\right)},X_{t+\gamma^{\prime }-1}^{\left(p;1\right)}). \end{array}$$
Let us now consider the window in $X^{p}$ of size $\lambda^{\prime }$ with a lag $\gamma_{pq}$ set to $-\gamma_{K-1}$. Using the same reasoning as before, one obtains:
(5)$$\begin{array}{cc} I(X_{t}^{\left(q;\lambda^{\prime }\right)};X_{t-\gamma_{K-1}}^{\left(p;\lambda^{\prime }\right)}|X_{t-1}^{\left(q;1\right)},X_{t-\gamma_{K-1}-1}^{\left(p;1\right)}) & =I(X_{t}^{\left(q;\gamma^{\prime }+\gamma_{K-1}\right)};X_{t-\gamma_{K-1}}^{\left(p;\lambda^{\prime }\right)}|X_{t-1}^{\left(q;1\right)},X_{t-\gamma_{K-1}-1}^{\left(p;1\right)})\\ \; & \;+I(X_{t+\gamma^{\prime }+\gamma_{K-1}}^{\left(q;\lambda^{\prime }-\gamma^{\prime }-\gamma_{K-1}\right)};X_{t-\gamma_{K-1}}^{\left(p;\lambda^{\prime }\right)}|X_{t+\gamma^{\prime }+\gamma_{K-1}-1}^{\left(q;1\right)},X_{t-\gamma_{K-1}-1}^{\left(p;1\right)}), \end{array}$$
with:
$$\begin{array}{cc} \; & I(X_{t+\gamma^{\prime }+\gamma_{K-1}}^{\left(q;\lambda^{\prime }-\gamma^{\prime }-\gamma_{K-1}\right)};X_{t-\gamma_{K-1}}^{\left(p;\lambda^{\prime }\right)}|X_{t+\gamma^{\prime }+\gamma_{K-1}-1}^{\left(q;1\right)},X_{t-\gamma_{K-1}-1}^{\left(p;1\right)})\\ \; & \geq I(X_{t+\gamma^{\prime }+\gamma_{K-1}}^{\left(q;\lambda^{\prime }-\gamma^{\prime }-\gamma_{K-1}\right)};X_{t+\gamma^{\prime }}^{\left(p;\min\left(\lambda ,\lambda^{\prime }-\gamma_{K-1}-\gamma^{\prime }\right)\right)}|X_{t+\gamma^{\prime }+\gamma_{K-1}-1}^{\left(q;1\right)},X_{t+\gamma^{\prime }-1}^{\left(p;1\right)}), \end{array}$$
as the window $X_{t-\gamma_{K-1}}^{\left(p;\lambda^{\prime }\right)}$ contains the window $X_{t+\gamma^{\prime }}^{\left(p;\min\left(\lambda ,\lambda^{\prime }-\gamma_{K-1}-\gamma^{\prime }\right)\right)}$. In addition, the first term on the right-hand side of Equation (Equation 5) is strictly positive as all the elements in $X_{t}^{\left(q;\gamma^{\prime }+\gamma_{K-1}\right)}$ have causal relations in $X_{t-\gamma_{K-1}}^{\left(p;\lambda^{\prime }\right)}$. Thus, the mutual information obtained with the negative lag $-\gamma_{K-1}$ is better than the one obtained with any positive lag:
$$I\left(X_{t}^{\left(q;\lambda^{\prime }\right)};X_{t-\gamma_{K-1}}^{\left(p;\lambda^{\prime }\right)}|X_{t-1}^{\left(q;1\right)},X_{t-\gamma_{K-1}-1}^{\left(p;1\right)}\right)>I\left(X_{t}^{\left(q;\lambda^{\prime }\right)};X_{t+\gamma^{\prime }}^{\left(p;\lambda \right)}|X_{t-1}^{\left(q;1\right)},X_{t+\gamma^{\prime }-1}^{\left(p;1\right)}\right);$$
meaning that the optimal lag given by CTMI will necessarily go from $X^{p}$ to $X^{q}$, which corresponds to the correct orientation.  □

The following orientation rule is a direct application of the above property.

**ER-rule** **0**(Entropy reduction—γ)**.**
*In a pair $X^{p}-X^{q}$, such $X^{p}$ and $X^{q}$ do not have a possible spurious correlation, if ${\bar{\gamma }}_{pq}>0$, then orient the edge as: $X^{p}\to X^{q}$.*

Furthermore, we make use of the following rule to orient additional edges when the optimal lag ${\bar{\gamma }}_{pq}$ is null based on the fact that CTMI increases asymmetrically with respect to the increase of $\lambda_{pq}$ and $\lambda_{qp}$ (Figure 3). This rule infers the direction of the cause by checking the difference in the window sizes as the window size of the cause cannot be greater than the window size of the effect.

**ER-rule** **1**(Entropy reduction—λ)**.**
*In a pair $X^{p}-X^{q}$, such $X^{p}$ and $X^{q}$ do not have a possible spurious correlation, if ${\bar{\gamma }}_{pq}=0$ and ${\bar{\lambda }}_{pq}<{\bar{\lambda }}_{qp}$, then orient the edge as: $X^{p}\to X^{q}$.*

In practice, we also apply the ER-rule 0 before PC-rules, because, experimentally, we found that the ER-rule 0 is more reliable than the PC-rule 0 in detecting lagged unshielded colliders, especially in the case of a low sample size.

We call our method PCTMI; the pseudo-code is available in Algorithm 1. Adj(Xq,G) represents all adjacent nodes to Xq in the graph G, and sepset(p,q) is the separation set of Xp and Xq. The output of the algorithm is a CPDAG version of the summary graph such that all lagged relations are oriented, but instantaneous relations are partially oriented.
**Algorithm 1**PCTMI.*X* a *d*-dimensional time series of length *T*, γmax∈N the maximum number of lags, α a significance thresholdForm a complete undirected graph G=(V,E) with *d* nodesn = 0**while** there exists Xq∈V such that card(Adj(Xq,G))≥n+1 **do** D=list() **for** Xq∈V s.t. card(Adj(Xq,G))≥n+1 **do**   **for** Xp∈Adj(Xq,G) **do**    **for** all subsets XR⊂Adj(Xq,G)\{Xp} such that card(XR)=n and (Γrp≥0 or Γrq≥0) for all r∈R **do**    yq,p,R=CTMI(Xp;Xq∣XR)    append (D,{Xq,Xp,XR,yq,p,R})) Sort D by increasing order of *y* **while** D is not empty **do**   {Xq,Xp,XR,y}=pop(D)   **if** Xp∈Adj(Xq,G) and XR⊂Adj(Xq,G) **then**    Compute *z* the p-value of CTMI(Xp;Xq∣XR) given by Equation (Equation 4)    **if** test z>α **then**    Remove edge Xp−Xq from G    Sepset(p,q)=Sepset(q,p)=XR n = n + 1**for** each triple in G, **do** apply PC-rule 0**while** no more edges can be oriented **do** **for** each triple in G, **do** apply PC-rules 1, 2, and 3**for** each connected pair in G **do** apply ER-rules 0 and 1**Return**
G

### 4.2. Extension to Hidden Common Causes

When unobserved variables are causing a variable of interest, the PC algorithm is no longer appropriate and one needs to resort to the FCI algorithm introduced in [9], which infers a partial ancestral graph (PAG). We extend here the version of this algorithm presented in [29] without the selection bias.

We recall the notations needed for FCI: a double arrow ←→ indicates the presence of hidden common causes, while the symbol ∘ represents an undetermined mark, and the meta-symbol asterisk * is used as a wildcard, which can denote a tail, an arrow, or a circle. Furthermore, we make use of the notion *Possible-Dsep*, which is defined as follows:

**Definition** **10**(Possible-Dsep [9])**.**
*$X^{r}$ is in the Possible-Dsep set of $X^{p}$ and $X^{q}$ if and only if $X^{r}$ is different from $X^{p}$ and $X^{q}$ and there is an undirected path U between $X^{p}$ and $X^{r}$ such that, for every subpath $<X^{w},X^{s},X^{v}>$ of U, either $X^{s}$ is a collider on the subpath or $X^{w}$ and $X^{v}$ are adjacent in the PAG.*

Lastly, we make use of the following types of paths. A *discriminating path* between $X^{p}$ and $X^{q}$ is a path that includes at least three edges such that each non-endpoint vertex $X^{r}$ on the path is adjacent to $X^{q}$, and $X^{p}$ is not adjacent to $X^{q}$, and every vertex between $X^{p}$ and $X^{r}$ is a collider on the path and is a parent of $X^{q}$. An *uncovered path* is a path where every consecutive triple on the path is unshielded. A *potentially directed path* is a path where the edge between two consecutive vertices is not directed toward the first or against the second.

From the skeleton obtained in Section 4.1.1, unshielded colliders are detected using the FCI-rule 0.

**FCI-rule** **0**(Origin of causality)**.**
*For each unshielded triple $X^{p}*\;\;-\;\;\circ X^{r}\circ \;\;-\;\;*X^{q}$, if $X^{r}\notin Sepset\left(p,q\right)$, then orient the unshielded triple as a collider: $X^{p}*\;\;\;\to X^{r}\leftarrow \;\;\;*X^{q}$.*

From this, Possible-Dsep sets can be constructed. As elements of Possible-Dsep sets in a PAG play a role similar to the ones of parents in a DAG, additional edges are removed by conditioning on the elements of the Possible-Dsep sets, using the same strategy as the one given in Section 4.1.1. All edges are then unoriented, and the FCI-rule 0 is again applied as some of the edges of the unshielded colliders originally detected may have been removed by the previous step. Then, as in FCI, FCI-rules 1, 2, 3, 4, 8, 9, and 10 are applied.

**FCI-rule** **1.**
*In an unshielded triple $X^{p}*\;\;\;\to X^{r}\circ \;\;-\;\;*X^{q}$, if $X^{r}\in Sepset\left(p,q\right)$, then orient the unshielded triple as $X^{p}*\;\;\;\to X^{r}*\;\;\;\to X^{q}$.*


**FCI-rule** **2.**
*If there exists a triple $X^{p}\to X^{r}*\;\;\;\to X^{q}$ or a triple $X^{p}*\;\;\;\to X^{r}\to X^{q}$ with $X^{p}*\;\;-\;\;\circ X^{q}$, then orient the pair as $X^{p}*\;\;\;\to X^{q}$.*


**FCI-rule** **3.**
*If there exists an unshielded triple $X^{p}*\;\;\;\to X^{r}\leftarrow \;\;\;*X^{q}$ and an unshielded triple $X^{p}*\;\;-\;\;\circ X^{s}\circ \;\;-\;\;*X^{q}$ and $X^{s}*\;\;-\;\;\circ X^{r}$, then orient the pair as $X^{s}\to X^{r}$.*


**FCI-rule** **4.**
*If there exists a discriminating path between $X^{p}$ and $X^{q}$ for $X^{r}$ and $X^{r}\circ \;\;-\;\;*X^{q}$, then orient $X^{r}\circ \;\;-\;\;*X^{q}$ as $X^{r}\to X^{q}$; otherwise, orient the triple as $X^{s}\leftarrow \;\;\;\to X^{r}\leftarrow \;\;\;\to X^{q}$.*


**FCI-rule** **8.**
*If $X^{p}\to X^{r}\to X^{q}$ or $X^{p}-\;\circ X^{r}\to X^{q}$ and $X^{p}\circ \;\;\;\to X^{q}$, then orient $X^{p}\to X^{q}$.*


**FCI-rule** **9.**
*If $X^{p}\circ \;\;\;\to X^{q}$ and U is an uncovered potentially directed path from $X^{p}$ to $X^{q}$ such that $X^{q}$ and $X^{r}$ are not adjacent, then orient the pair as $X^{p}\to X^{q}$.*


**FCI-rule** **10.**
*Suppose $X^{p}\circ \;\;\;\to X^{q}$, $X^{r}\to X^{q}\leftarrow X^{s}$, $U_{1}$ is an uncovered potentially directed path from $X^{p}$ to $X^{r}$, and $U_{2}$ is an uncovered potentially directed path from $X^{p}$ to $X^{s}$. Let μ be the vertex adjacent to $X^{p}$ on $U_{1}$ (μ could be $X^{r}$) and ω be the vertex adjacent to $X^{p}$ on $U_{2}$ (ω could be $X^{s}$). If μ and ω are distinct and are not adjacent, then orient $X^{p}\circ \;\;\;\to X^{q}$ as $X^{p}\to X^{q}$.*


Note that we have not included FCI-rules 5, 6, and 7 from [29], as these rules deal with selection bias, a phenomenon that is not present in the datasets we consider. Including these rules in our framework is nevertheless straightforward.

Finally, as in PCTMI, we orient additional edges using an adapted version of the ER-rules.

**LER-rule** **0**(Latent entropy reduction—γ)**.**
*In a pair $X^{p}*\;\;-\;\;\circ X^{q}$, such $X^{p}$ and $X^{q}$ do not have a possible spurious correlation, if ${\bar{\gamma }}_{pq}>0$, then orient the edge as: $X^{p}*\;\;\;\to X^{q}$.*

**LER-rule** **1**(Latent entropy reduction—λ)**.**
*In a pair $X^{p}*\;\;-\;\;\circ X^{q}$, such $X^{p}$ and $X^{q}$ do not have a possible spurious correlation, if ${\bar{\gamma }}_{pq}=0$ and ${\bar{\lambda }}_{pq}<{\bar{\lambda }}_{qp}$, then orient the edge as: $X^{p}*\;\;\;\to X^{q}$.*

The overall process, referred to as FCITMI, is described in Algorithm 2.
**Algorithm 2**FCITMI.
**Require:***X* a *d*-dimensional time series of length *T*, γmax∈N the maximum number of lags, α a significance thresholdForm a complete undirected graph G=(V,E) with *d* nodesn = 0**while** there exists Xq∈V such that card(Adj(Xq,G))≥n+1 **do** D=list() **for** Xq∈V s.t. card(Adj(Xq,G))≥n+1 **do**   **for** Xp∈Adj(Xq,G) **do**    **for** all subsets XR⊂Adj(Xq,G)\{Xp} such that card(XR)=n and (γrp≥0 or γrq≥0) for all r∈R **do**    yq,p,R=CTMI(Xp;Xq∣XR)    append (D,{Xq,Xp,XR,yq,p,R})) Sort D by increasing order of *y* **while** D is not empty **do**   {Xq,Xp,XR,y}=pop(D)   **if** Xp∈Adj(Xq,G) and XR⊂Adj(Xq,G) **then**    Compute *z* the p-value of CTMI(Xp;Xq∣XR) given by Equation (Equation 4)    **if** test z>α **then**    Remove edge Xp−Xq from G    Sepset(p,q)=Sepset(q,p)=XR n = n + 1**for** each triple in G, **do** apply FCI-rule 0using Possible-Dsep sets, remove edges using CTMIReorient all edges as ∘−∘ in G**for** each triple in G, **do** apply FCI-rule 0**while** edges can be oriented **do** **for** each triple in G, apply FCI-rules 1, 2, 3, 4, 8, 9, and 10 **for** each connected pair in G, **do** apply ER-rules 0 and 1.**Return**
G

## 5. Experiments

In this section, we evaluate our method experimentally on several datasets. We propose first an extensive analysis on simulated data with equal and different sampling rates, generated from basic causal structures; then, we performed an analysis on real-world datasets. To assess the quality of causal inference, we used the F1-score regarding directed edges in the graph without considering self-loops.

In the following, we first describe the different settings of the methods we compare with and the datasets, and then, we describe the results and provide a complexity analysis.

### 5.1. Methods and Their Use

We compared our method PCTMI, available at https://github.com/ckassaad/PCTMI (accessed on 29 June 2022),with several methods. From the Granger family, we ran MVGC through the implementation available at http://www.sussex.ac.uk/sackler/mvgc/ (accessed on 29 June 2022) and TCDF through the implementation available at https://github.com/M-Nauta/TCDF (accessed on 29 June 2022). For TCDF, some hyperparameters have to be defined: we use da kernel of size 4, a dilation coefficient 4, one hidden layer, a learning rate of 0.01, and 5000 epochs. From the score-based family, we retained Dynotears, which is available at https://github.com/quantumblacklabs/causalnex (accessed on 29 June 2022), the hyperparameters of which were set to their recommended values (the regularization constants $\lambda_{W}=\lambda_{A}=0.05$). Among the noise-based approaches, we ran TiMINo, which is available at http://web.math.ku.dk/~peters/code.html (accessed on 29 June 2022). TiMINo uses a test based on the cross-correlation.From the constraint-based family, we ran PCMCI using the mutual information to measure the dependence, both provided in the implementation available at https://github.com/jakobrunge/tigramite (accessed on 29 June 2022).

We compared FCITMI with tsFCI, provided at https://sites.google.com/site/dorisentner/publications/tsfci (accessed on 29 June 2022), where independence or conditional independence is tested, respectively, with tests of zero correlation or zero partial correlation.

For all methods using mutual information, we used the k-nearest neighbor estimator [26], for which we fixed the number of nearest neighbor to k=10. Since the output of those measures are necessarily positive given a finite sample size and a finite numerical precision, we used a significance permutation test introduced in [27]. For all methods, we used $\gamma_{\max}=5$, and when performing a statistical test, we used a significance level of 0.05.

### 5.2. Datasets

To illustrate the behavior of the causal inference algorithms, we relied on both artificial and real-world datasets.

#### 5.2.1. Simulated data

In the case of causal sufficiency, we simulated time series with equal and different sampling rates of size 1000 generated from three different causal structures: fork, v structure, and diamond represented in Table 1. In the case of non-causal sufficiency, we considered the 7ts2h structure represented in Table 2.

For each structure, we generated 10 datasets using the following generating process: for all *q*, $X_{0}^{q}=0$, and for all t>0,
Xtq=at−1qqXt−1q+∑(p,γ)Xt−γp∈Par(Xtq)at−γpqf(Xt−γp)+0.1ξtq,
where γ≥0, $a_{t}^{jq}$ are random coefficients chosen uniformly in U([−1;−0.1]∪[0.1;1]) for all 1≤j≤d, $\xi_{t}^{q}∼N\left(0,\sqrt{15}\right)$ and *f* is a nonlinear function chosen at random uniformly among the absolute value, tanh, sine, and cosine.

#### 5.2.2. Real Data

First, we considered the realistic functional magnetic resonance imaging (fMRI) benchmark, which contains blood-oxygen-level dependent (BOLD) datasets [30] for 28 different underlying brain networks. It measures the neural activity of different regions of interest in the brain based on the change of blood flow. There are 50 regions in total, each with its own associated time series. Since not all existing methods can handle 50 time series, datasets with more than 10 time series were excluded. In total, we were left with 26 datasets containing between 5 and 10 brain regions.

Second, we considered time series collected from an IT monitoring system with a one-minute sampling rate provided by EasyVista (https://www.easyvista.com/fr/produits/servicenav, accessed on 29 June 2022). In total, we considered 24 datasets with 1000 timestamps. Each dataset contains three time series: the *metric extraction* (M.E), which represents the activity of the extraction of the metrics from the messages; the *group history insertion* (G.H.I), which represents the activity of the insertion of the historical status in the database; and the *collector monitoring information* (C.M.I), which represents the activity of the updates in a given database. Lags between time series are unknown, as well as the existence of self-causes. According to the domain experts, all these datasets follow a fork structure such that *metric extraction* is a common cause of the other two time series.

### 5.3. Numerical Results

#### 5.3.1. Simulated Data

We provide in Table 3 the performance of all methods on simulated data with an equal sampling rate. PCTMI has better results than other methods for the v structure and the fork structure For the diamond structure, PCTMI and PCMCI have the same performance; however, PCTMI has a lower standard deviation. In general, we can say, that constraint-based algorithms perform best.

We also assess the behavior of PCTMI when the time series have different sampling rates in Table 4. We present here results only for PCTMI, because other methods are not applicable to time series with different sampling rates. As one can see, the performance obtained here is close to the ones obtained with equal sampling rates in the case of the v structure, but significantly lower in the case if the fork structure and the diamond. The degradation of the results is not really surprising as one has less data to rely on in a different sampling rate scenario. By comparing the results of PCTMI on time series with different sampling rates with the results of other methods obtained when time series have an equal sampling rate, we can see that PCTMI still performs better than most methods.

In the case of causal non-sufficiency, our proposed algorithm FCITMI outperforms tsFCI, as shown in Table 5.

#### 5.3.2. Real Data

We provide in Table 6 the results on the fMRI benchmark and the IT benchmark for all the methods. In the case of fMRI, VarLiNGAM performs best, followed by Dynotears and, then, by PCTMI and TiMINo. The success of linear methods (VarLiNGAM and Dynotears) on this benchmark might suggest that causal relations in fMRI are linear, especially as this result was not replicated on other datasets considered in this paper. In the case of the IT datasets, TiMINo performs best, followed by PCTMI; however, while investigating the results, we noticed that for 6/10 datasets, TiMINo returns a fully connected summary graph with arcs oriented from both sides. In both benchmarks, PCTMI clearly outperforms other constraint-based methods.

In the case of the IT datasets, we also provide in Table 7, for each dataset, the inferred summary causal graphs by methods that have an F1-score >0 in Table 6. Here, we can see that in 5 out of 10 datasets, PCTMI was able to infer that M.E is a cause of G.H.I. On the other hand, PCTMI was able to infer that M.E is a cause of C.M.I. in only one dataset. Interestingly, in 9 out 10 datasets, PCTMI did not infer any false positive. As one can expect from Table 6, PCMCI suffers on all datasets; however, one can notice that as PCTMI, PCMCI has a tendency to yield sparse graphs. TiMINo, which has the best F1-score in Table 6, seems to have a very low precision. In 7 out 10 datasets, TiMINo returned a complete bidirected graph. In the other three datasets, TiMINo detects all true causal relations, in addition to one false positive. VarLiNGAM has an even lower precision compared to TiMINo and a lower recall. Finally, MVGC infers most of the time a complete bidirected graph.

#### 5.3.3. Complexity Analysis

Here, we provide a complexity analysis on different constraint-based algorithms including our proposed method.

In the worst case, the complexity of PC in a window causal graph is bounded by $\frac{{\left(d\gamma_{max}\right)}^{2}{\left(d\gamma_{max}-1\right)}^{k-1}}{(k-1)!}$, where *k* represents the maximal degree of any vertex and $\gamma_{max}$ is the maximum number of lags. Each operation consists of conducting a significance test on a conditional independence measure. Algorithms adapted to time series, as PCMCI [7], rely on time information to reduce the number of tests. PCTMI reduces the number of tests even more since it infers a summary graph. Indeed, PCTMI’s complexity in the worst case is bounded by $\frac{d^{2}{\left(d-1\right)}^{k-1}}{(k-1)!}$.

Figure 6 provides an empirical illustration of the difference in the complexity of the two approaches on the three structures (v structure, fork, diamond), sorted according to their number of nodes, their maximal out-degree, and their maximal in-degree. The time is given in seconds. As one can note, PCTMI is always faster than PCMCI, the difference being more important when the structure to be inferred is complex.

#### 5.3.4. Hyperparameters’ Analysis

Here, we provide a hyperparameter analysis on CTMI with respect to the maximum lag $\gamma_{\max}$ and the k-nearest neighbor *k*. Using the same 10 datasets that are compatible with the fork structure in Table 1 and generated as described in Section 5.2.1, we performed two experimentations.

First, we computed the average and the standard deviation of CTMI($X^{2};X^{3}$) and CTMI($X^{2};X^{3}|X^{1}$) over the 10 datasets while varying $\gamma_{\max}$ between 3 and 10. The results given in Figure 7a show that CTMI is robust with respect to $\gamma_{\max}$ in the case of dependency and in the case of conditional independency: the mean of CTMI($X^{2};X^{3}$) remains the same for $\gamma_{\max}=3,4,5$ and slightly decreases for $\gamma_{\max}=10$, and the standard deviation remains the same for $\gamma_{\max}=3,4$ and slightly increases for $\gamma_{\max}=5,10$. On the other hand, the mean and the standard deviation of CTMI($X^{2};X^{3}|X^{1}$) remain the same for all $\gamma_{\max}$.

Second, we again computed the average and the standard deviation of CTMI($X^{2};X^{3}$) and CTMI($X^{2};X^{3}|X^{1}$) over the 10 datasets, but now, while varying *k* between 5 and 100. The results given in Figure 7b show that CTMI is robust with respect to *k* in the case of conditional independency, but suffers in the case of dependency when *k* increases: the mean of CTMI($X^{2};X^{3}$) slightly decreases between k=5 and k=10, but drastically decreases for k=50.

## 6. Discussion and Conclusions

We addressed in this paper the problem of learning a summary causal graph on time series with equal or different sampling rates. To do so, we first proposed a new temporal mutual information measure defined on a window-based representation of time series. We then showed how this measure relates to an entropy reduction principle, which can be seen as a special case of the probability raising principle. We finally combined these two ingredients in PC-like and FCI-like algorithms to construct the summary causal graph. Our method proved to work well with small time complexity in comparison with similar approaches that use mutual information. The main limitations of our methods are that they cannot orient causal relations that are not oriented by the classical PC-rules or FCI-rules and have a possible spurious correlation, and they assume acyclicity in summary causal graphs.

## Figures and Tables

**Figure 1 entropy-24-01156-f001:**
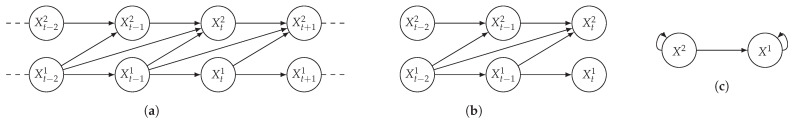
Example of a full-time causal graph (**a**), a window causal graph (**b**), and a summary causal graph (**c**).

**Figure 2 entropy-24-01156-f002:**
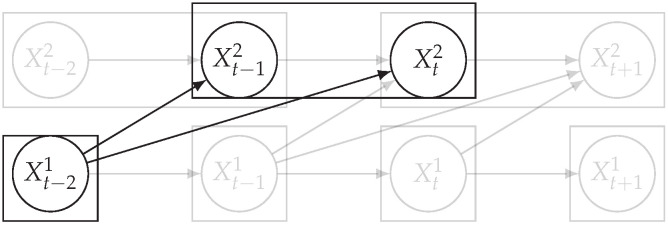
Why do we need windows and lags? An illustration with two time series where $X^{1}$ causes $X^{2}$ in two steps (circles correspond to observed points and rectangles to windows). The arrows in black are discussed in the text.

**Figure 3 entropy-24-01156-f003:**
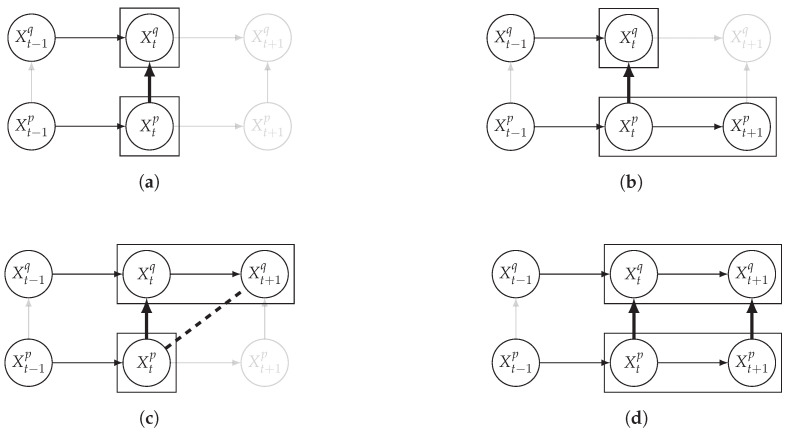
Illustration of the asymmetric increase of CTMI with the increase of the window sizes. The mutual information conditioned on the past in (**a**) is $I(X_{t}^{p},X_{t}^{q}|X_{t-1}^{p},X_{t-1}^{q})>0$. It increases when increasing only the window size of the effect, as in (**c**), i.e, $I\left(X_{t}^{p},X_{t}^{\left(q;2\right)}|X_{t-1}^{p},X_{t-1}^{q}\right)>I\left(X_{t}^{p},X_{t}^{q}|X_{t-1}^{p},X_{t-1}^{q}\right)$, or when increasing simultaneously the window sizes of the effect and the cause, as in (**d**), i.e, $I\left(X_{t}^{\left(p;2\right)},X_{t}^{\left(q;2\right)}|X_{t-1}^{p},X_{t-1}^{q}\right)>I\left(X_{t}^{p},X_{t}^{\left(q;2\right)}|X_{t-1}^{p},X_{t-1}^{q}\right)$. However, it does not increase when increasing only the window size of the cause, as in (**b**), i.e, $I\left(X_{t}^{\left(p;2\right)},X_{t}^{q}|X_{t-1}^{p},X_{t-1}^{q}\right)=I\left(X_{t}^{p},X_{t}^{q}|X_{t-1}^{p},X_{t-1}^{q}\right)$. Dashed lines are for correlations that are not causations, and bold arrows correspond to causal relations between the window representations of time series.

**Figure 4 entropy-24-01156-f004:**
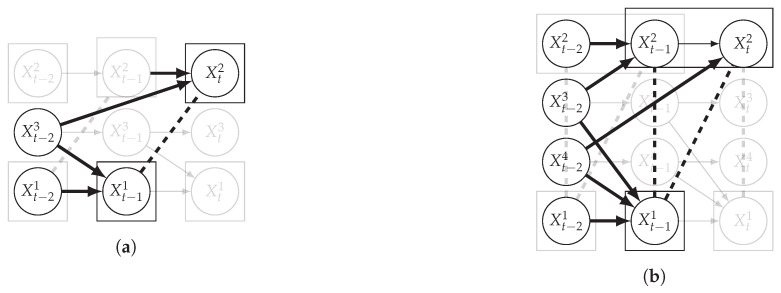
Example of conditional independence between dependent time series. In (**a**) the conditioning set contains one time series $X^{3}$ in addition to the past of $X^{1}$ and $X^{2}$. In (**b**) the conditioning set contains one time series $X^{3}$ and $X^{4}$ in addition to the past of $X^{1}$ and $X^{2}$. Dashed lines are for correlations that are not causations, and bold arrows correspond to conditioning variables.

**Figure 5 entropy-24-01156-f005:**
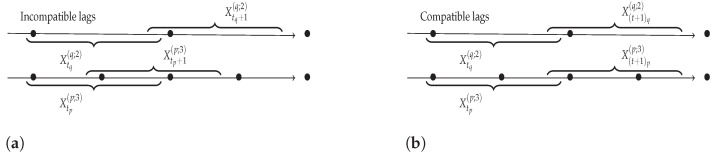
Illustration for constructing sequences of windows for two time series with different sampling rates. In (**a**) the construction represents incompatible lags and in (**b**) the construction represents compatible lags.

**Figure 6 entropy-24-01156-f006:**
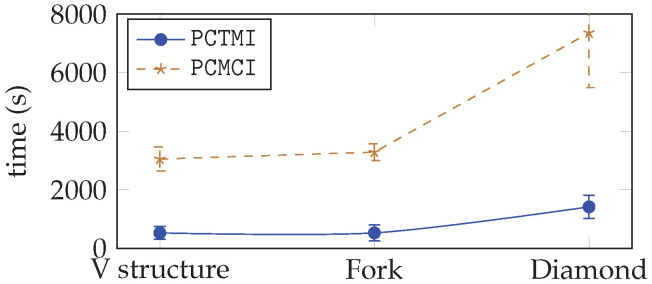
Time computation for PCTMI and PCMCI.

**Figure 7 entropy-24-01156-f007:**
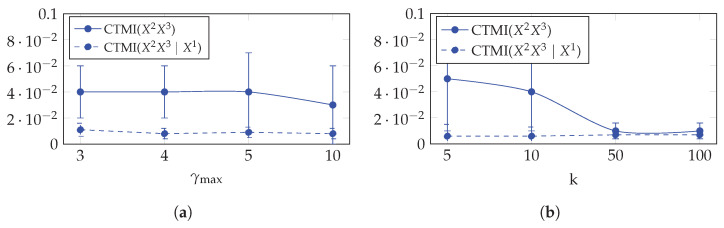
CTMI with respect to the maximum lag $\gamma_{\max}$ in (**a**) and the k-nearest neighbor *k* in (**b**) for dependent time series ($X^{2}$ and $X^{3}$ in the fork structure in Table 1) and conditionally independent time series ($X^{2}$ and $X^{3}$ conditioned on $X^{1}$ in the fork structure in Table 1).

**Table 1 entropy-24-01156-t001:** Structures of simulated data without hidden common causes. A→B means that *A* causes *B*.

V-Structure	Fork	Diamond
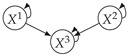	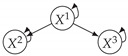	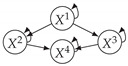

**Table 2 entropy-24-01156-t002:** Structures of simulated data with hidden common causes. A→B means that A causes B and A←→B represents the existence of a hidden common cause between A and B.

7ts2h
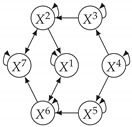

**Table 3 entropy-24-01156-t003:** Results for simulated datasets with an equal sampling rate. We report the mean and the standard deviation of the F1-score. The best results are in bold.

	PCTMI	PCMCI	TiMINo	VarLiNGAM	Dynotears	TCDF	MVGC
V structure	0.78±0.18	0.67±0.37	0.65±0.37	0.0±0.0	0.07±0.20	0.13±0.26	0.37±0.26
Fork	0.83±0.31	0.78±0.17	0.52±0.44	0.0±0.0	0.07±0.20	0.26±0.32	0.44±0.38
Diamond	0.82±0.11	0.82±0.16	0.60±0.25	0.03±0.09	0.23±0.24	0.16±0.19	0.68±0.26

**Table 4 entropy-24-01156-t004:** Results for simulated datasets with different sampling rates. We report the mean and the standard deviation of the F1-score.

	PCTMI
V structure	0.80±0.31
Fork	0.56±0.30
Diamond	0.66±0.24

**Table 5 entropy-24-01156-t005:** Results for simulated datasets with an equal sampling rate and with hidden common causes. We report the mean and the standard deviation of the F1-score.

	FCITMI	tsFCI
7ts2h	0.44±0.11	0.37±0.09

**Table 6 entropy-24-01156-t006:** Results for real datasets. We report the mean and the standard deviation of the F1-score. The best results are in bold.

	PCTMI	PCMCI	TiMINo	VarLiNGAM	Dynotears	TCDF	MVGC
fMRI	0.32±0.17	0.22±0.18	0.32±0.11	0.49±0.28	0.34±0.13	0.07±0.13	0.24±0.18
IT	0.40±0.32	0.25±0.31	0.62±0.14	0.36±0.19	0.0±0.0	0.0±0.0	0.38±0.17

**Table 7 entropy-24-01156-t007:** Summary causal graphs inferred by different methods using 10 different monitoring IT datasets. A→B means that *A* causes *B* and A⇄B means that *A* causes *B* and *B* causes *A*.

	PCTMI	PCMCI	TiMINo	VarLiNGAM	MVGC
Dataset 1	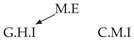	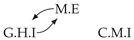	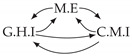	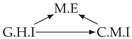	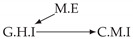
Dataset 2	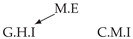	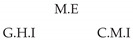	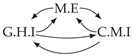	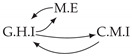	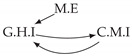
Dataset 3	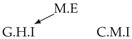	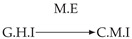	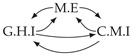	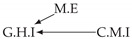	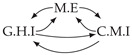
Dataset 4	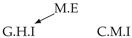	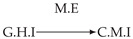	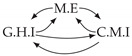	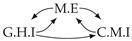	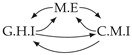
Dataset 5	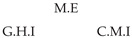	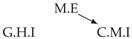	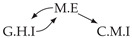	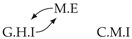	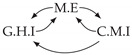
Dataset 6	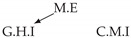	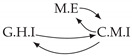	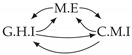	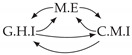	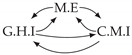
Dataset 7	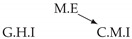	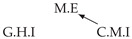	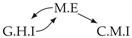	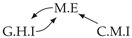	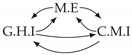
Dataset 8	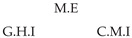	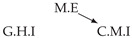	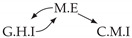	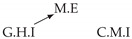	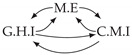
Dataset 9	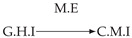	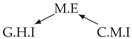	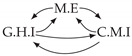	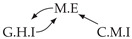	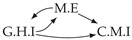
Dataset 10	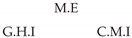	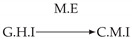	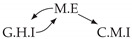	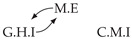	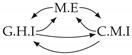

## Data Availability

All data presented in this study are publicly available: Simulated data are available at https://dataverse.harvard.edu/dataverse/basic_causal_structures_additive_noise (accessed on 29 June 2022); The original fMRI data are available at https://www.fmrib.ox.ac.uk/datasets/netsim/index.html (accessed on 29 June 2022), and a preprocessed version is available at https://github.com/M-Nauta/TCDF/tree/master/data/fMRI (accessed on 29 June 2022); IT monitoring data are available at https://easyvista2015-my.sharepoint.com/:f:/g/personal/aait-bachir_easyvista_com/ElLiNpfCkO1JgglQcrBPP9IBxBXzaINrM5f0ILz6wbgoEQ?e=OBTsUY (accessed on 29 June 2022).

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
