# Peer review of "Entropy-Based Discovery of Summary Causal Graphs in Time Series"

_entropy, 2022, doi:10.3390/e24081156_

Round 1

Reviewer 1 Report

 I can not find the  definition of the  concept of  summary causal  graph in this paper,  and the example  shown in Figure 1 (c)  make me more confused.  As a result, it is hard for me to assess  the proposed methods. 

 The paper just mentioned a few papers  that deal with time series or process data.  There are many causal structure learning methods omitted, such as   dynamic Bayesian network,  CTBN,  multivariate hawks processes,  SVAR etc.

Reviewer 2 Report

The authors proposed a new causal measure, CTMI, considering time lag and time window, which is analogous to transfer entropy. They show CTMI is a measure in a sense of Wiener's principle of causality. They also defined conditional version of CTMI. The KNN estimators of the CTMI is proposed. The PC and FCI algorithms based on CTMI are also proposed. The proposed PC algorithm was compared with 6 similar ones in both simulated and real data experiments.

Unlike transfer entropy, the physical meaning of the definition of (1) is unclear. Why the MAXIMUM, instead of other statistics, of the standard MIs, is considered here? Why conditional on the past with window size=1? The author should give more explanations on this definition. It would be better if a real example is presented to show the definition is reasonable.

The experiment is somewhat weak. I suggest that The basic PC and FCI should be included in the comparison, which are implemented in many OSS, such as the R package {bnlearn} or {pcalg}. The hyper-parameter for estimating CTMI is fixed in the experiments. Considering the importance of hyperparameter for CTMI, more hyperparameter tuning should be done to know the performance of CTMI. The second real data from IT monitoring is somewhat simple for testing the performance of the algorithms on causal structure learning. The experimental results would be presented much better and clear if with more figures on the estimated graph.

There are many information theoretical-based measures between time series related to current work, such as the famous Directed Information, or causation entropy (Sun, Taylor, and Bollt, 2015), which are suggested to discuss in the related work.

Others:

'FMRI' should be 'fMRI'.

Reviewer 3 Report

The manuscript is well written and carefully organised. All the relevant topics are well introduced and adequately discussed.

The main issue I have about the manuscript refers to the innovative content. What the authors called the temporal mutual information is nothing else as the transfer entropy, which was originally proposed in the year 2000 and is now well known. The additional issue of time series having a different time index can be easily solved with linear interpolation and resampling. The authors should better prove what is the competitive advantage of their probably unnecessarily complex procedure.

The fact that the transfer entropy can improve PC-like and FCI-like algorithms is probably be worth documenting. I’ll leave tot h editor to decide whether the manuscript is sufficiently innovative to be published in the journal. However I would recommend to better acknowledge what is already known and done in the past.

Round 2

Reviewer 2 Report

Thank you for your response and revision according to remarks. I think this work has been largely improved. I agree to accept it for Entropy.
